# Evaluation of the Impact of Antibiogram Availability and Utilization on Antibiotic Use among Healthcare Providers in Saudi Arabia: A Cross-Sectional Study

**DOI:** 10.3390/microorganisms12071444

**Published:** 2024-07-16

**Authors:** Haytham A. Wali, Hassan Ali Alhajji, Ridha Alsaeed, Abdullah Aldughaim, Rakan Almutairi, Amira S. Radwan

**Affiliations:** Department of Pharmacy Practice, College of Clinical Pharmacy, King Faisal University, Al-Ahsa 36362, Saudi Arabia; 219020939@student.kfu.edu.sa (H.A.A.); aradwan@kfu.edu.sa (A.S.R.)

**Keywords:** drug resistance, microbial sensitivity tests, antimicrobial stewardship, health personnel, cross-sectional studies

## Abstract

Antibiotic resistance is a significant global health issue, exacerbated by the indiscriminate use of antibiotics without antibiograms. Implementing appropriate stewardship programs that monitor and control antibiotic use is essential to minimize resistance development and ensure optimal patient outcomes. This study aims to assess the impact of antibiogram availability and utilization on antibiotic use among healthcare providers in Saudi Arabia, focusing on whether antibiogram data and utilization influence the appropriateness of antibiotic prescribing practices. A cross-sectional study design was employed, utilizing a self-administered online survey distributed to physicians, pharmacists, and other healthcare providers across various healthcare settings in Saudi Arabia. Data were collected over a 90-day period, from 1 December 2023 to 29 February 2024. Descriptive statistics were used to summarize participants’ characteristics, and data were analyzed based on geographical region, participants’ positions, and other predetermined differences. Out of 23,860 contacted individuals, 333 responded, and 283 were included in the analysis. The majority (62.2%) reported the availability of antibiograms at their facilities, with 84.1% utilizing them in clinical practice. The frequency of consulting antibiograms varied, with only 21.6% doing so daily. Key barriers to antibiogram availability included lack of expertise, technological infrastructure, and funding. Most participants (68.8%) observed changes in antibiotic prescription practices post-antibiogram introduction, and 86.9% believed antibiograms could shorten patients’ length of stay and improve outcomes. However, only 40.9% had received training on antibiogram use. While healthcare providers in Saudi Arabia have a relatively high level of antibiogram availability and utilization, several barriers hinder their consistent use. Enhancing accessibility, promoting frequent use, and providing targeted training could strengthen the impact of antibiograms on antibiotic prescribing practices and antimicrobial stewardship efforts.

## 1. Introduction

Antibiotic resistance has become a global health concern, increasing morbidity, mortality, and healthcare costs [1]. The indiscriminate use of antibiotics without antibiograms can lead to the development of antibiotic-resistant bacteria and negatively impact patient outcomes [2]. Therefore, it is vital to implement appropriate stewardship programs that monitor and control antibiotic use to minimize the risk of resistance development and ensure optimal patient outcomes.

One strategy to promote appropriate antibiotic use is the availability and utilization of antibiograms, which provide local data on antibiotic susceptibility patterns [3,4]. A recent systematic review study conducted in the United States included 37 studies that found a positive impact of the use of antibiograms on antibiotic consumption, appropriateness of prescribing, and treatment costs. Antibiograms allow healthcare providers to make evidence-based decisions regarding antibiotic prescribing, improving patient outcomes, and decreasing antibiotic resistance [5].

The antibiogram offers a significant advantage over molecular platforms by providing a comprehensive, phenotypic assessment of a pathogen’s antibiotic resistance profile. While molecular platforms can rapidly identify the presence of specific resistance genes, they may miss novel or unexpected resistance mechanisms not covered by their assays [6]. In contrast, antibiograms are based on the actual growth of bacteria in the presence of various antibiotics, offering a direct measure of susceptibility and resistance [7]. This phenotypic approach not only detects resistance conferred by known genes but also captures resistance due to unknown or emerging mechanisms, ensuring more accurate and reliable guidance for effective antimicrobial therapy [8].

Antibiograms are invaluable tools in clinical practice for guiding the selection of appropriate antimicrobial therapy. By providing a localized summary of antimicrobial susceptibilities for prevalent bacterial pathogens, antibiograms enable healthcare providers to make informed decisions about empiric treatment before specific culture and sensitivity results are available [9]. This is particularly crucial in the context of rising antibiotic resistance, as it helps to optimize antibiotic use, reduce the risk of treatment failure, and minimize the development of further resistance [10]. Additionally, antibiograms support infection control efforts by identifying trends in resistance patterns, thereby aiding in formulating hospital antibiograms and stewardship policies that promote the judicious use of antibiotics [11].

A diverse population of health professionals, including microbiologists, pharmacists, infection control nurses, and epidemiologists, is essential for effectively utilizing antibiograms. While doctors play a crucial role in diagnosing and treating infections, the successful implementation and interpretation of antibiograms require a multidisciplinary approach. Microbiologists provide the expertise in identifying pathogens and determining their resistance patterns, while pharmacists offer critical insights into antimicrobial stewardship and the pharmacodynamics of antibiotic therapies. Infection control nurses are pivotal in monitoring and preventing the spread of resistant organisms within healthcare settings, ensuring that the data from antibiograms are applied to improve infection control practices. Epidemiologists contribute by analyzing resistance trends and guiding public health interventions. This collaborative approach ensures that antibiograms are not only used to inform individual patient care but also to shape institutional policies and public health strategies, thereby optimizing antibiotic use and combating antimicrobial resistance more effectively.

This study aims to assess the impact of antibiogram availability and utilization on antibiotic use among healthcare providers in Saudi Arabia. Specifically, it seeks to determine whether antibiogram data and utilization influence the appropriateness of antibiotic prescribing practices.

## 2. Materials and Methods

### 2.1. Study Design

This study employed a cross-sectional design to evaluate the effect of antibiogram availability and utilization on antibiotic use among healthcare providers in Saudi Arabia. Data were collected through a survey distributed to physicians, pharmacists, and other healthcare providers across various healthcare settings, including hospitals, clinics, and primary care centers.

### 2.2. Subject Selection

This study included healthcare providers involved in prescribing and using antibiotics, including physicians and pharmacists. Non-healthcare providers, health administrators, maintenance staff, and those not directly involved in antibiotic prescriptions or working in non-clinical settings were excluded.

### 2.3. Instrumentation

This study utilized a self-administered survey distributed online via Google Forms (Google LLC, Menlo Park, CA, USA; available at: https://docs.google.com/forms/u/0/, access date: 29 February 2024). During the survey development, a content and face validity process was conducted to ensure that the survey was valid and relevant to the subject matter. This process involved soliciting feedback from five healthcare providers with relevant experiences in the field. The survey was carefully reviewed and evaluated, and comments and suggestions for improvements were provided.

Based on their feedback, the survey was modified to ensure that it accurately captured the relevant information and was easy for the study population to understand. The final version of the survey was then distributed to the study population to ensure that it accurately reflected the subject matter and met the needs of the research project.

### 2.4. Data Collection

Data collection spanned 90 days, from 1 December 2023 to 29 February 2024. Researchers emailed the survey to Saudi hospitals and healthcare providers, focusing on different regions sequentially: the Eastern Region in the first 30 days, the Central and Western Regions in the next 30 days, and the Northern and Southern Regions in the final 30 days. Reminders were sent to ensure that participants completed the survey. The online version of the survey was distributed using social media platforms such as Facebook (Meta Platforms, Inc., Cambridge, MA, USA, available at: https://www.facebook.com, access date: 29 February 2024), WhatsApp (Meta Platforms, Inc., Cambridge, MA, USA, available at: https://web.whatsapp.com, access date: 29 February 2024), and X (X Corp., San Francisco, CA, USA, available at: https://x.com/home, access date: 29 February 2024). The study investigators visited some hospitals in person and gave healthcare providers a QR code to complete the survey.

### 2.5. Data Analysis

Descriptive statistics were used to summarize the participants’ characteristics, including demographics, professional experience, and knowledge of antibiograms. Data were analyzed based on geographical region, participants’ positions, and other predetermined differences. The results are presented as frequencies and percentages and interpreted accordingly.

### 2.6. Ethical Clearance

This study adhered to ethical principles to protect participants’ rights and welfare. Approval from the King Faisal University Deanship of Scientific Research Institutional Review Board (IRB) was obtained prior to the commencement of this study (Ref. No. KFU-REC-2023-NOV-ETHICS1683). Informed consent was obtained from all subjects involved in this study through a mandatory question at the beginning of the survey asking about their agreement to participate.

## 3. Results

A total of 23,860 individuals were initially contacted to participate in this study, and 333 responded. Of these, 330 consented to participate in this study. After reviewing the responses, 47 participants were excluded owing to ineligibility. Finally, 283 responses were included in the analysis. The flow diagram in Figure 1 highlights the recruitment and retention processes of the study participants.

### 3.1. Demographic Information

This study included 283 participants (Table 1). The age distribution showed that 32.5% were under 30 years, 48.4% were between 31 and 40 years, 17% were 41–50 years, and 2.1% were 51–60 years. The sex distribution was 54.8% females. Regarding nationality, 70.3% were Saudi nationals, and 29.7% were non-Saudi residents. Participants were predominantly from the Eastern Region (45.2%), followed by the Central (28.3%), Western (14.8%), Southern (6.7%), and Northern (4.9%) Regions.

The participants’ job titles were mainly physicians (40.3%), pharmacists (33.6%), nurses (20.1%), and dentists (6%). Regarding specialties, 41.3% were non-specialized, 12% specialized in family medicine, 3.5% in infectious diseases, and the rest in various other specialties. The majority worked in Ministry of Health hospitals (44.5%) and private hospitals (39.2%). Experience levels varied: 29% had 6–10 years of experience, 27.2% had 11–20 years, and 25.4% had 1–5 years. Most participants worked in general hospitals (56.5%) or specialized hospitals (29.7%).

### 3.2. Evaluation of Antibiogram Availability

Among the participants, 62.2% reported the availability of antibiograms at their healthcare facilities, 16.6% reported unavailability, and 21.2% were unsure. The primary reasons for unavailability included lack of expertise (61.7%), lack of technological infrastructure and resources (57.4%), and lack of funding (44.7%). Among those aware of antibiogram availability, 41.1% found it very easy to access, while 41.7% found it moderately easy (Table 2).

### 3.3. Utilization of Antibiograms

Among the 176 participants with access to an antibiogram, 84.1% utilized it in their healthcare practice (Table 3). The frequency of referring to an antibiogram when prescribing antibiotics varied, with 21.6% consulting it daily, 25.6% weekly, 33.5% monthly, and 19.3% yearly. Key factors influencing the decision to consult an antibiogram included the patient’s clinical presentation (68.2%) and failure to respond to previously administered antibiotics (65.3%).

Updating knowledge about new antibiotics and resistance patterns through antibiograms was a regular practice for some, with 23.3% always doing so and 33.5% sometimes doing so. The frequency of updating the antibiogram by healthcare facility was mostly yearly (59.6%). The obstacles to using antibiograms included data quality issues (56.3%), delays in data updates (37.5%), and difficulty in data interpretation (36.9%). Approximately 40.9% had received training on utilizing antibiograms, and 46.6% received feedback on antibiotic prescribing practices based on antibiogram data.

### 3.4. Perception of the Impact of Antibiograms

Most participants (68.8%) observed changes in antibiotic prescription practices after the introduction of antibiograms (Table 4). The perceived effectiveness of antibiograms in reducing inappropriate antibiotic use and resistance varied, with 33% and 27.8% finding them effective and moderately effective, respectively. A significant majority (86.9%) believed that antibiograms could shorten patients’ length of stay and contribute to improved patient outcomes (84.7%). Additionally, 85.2% agreed that antibiograms lead to cost savings in healthcare settings.

The overall impact of antibiogram availability was rated positively, with 30.1% rating it as good, and 20.5% as very good. The confidence in interpreting and applying antibiogram data varied, with 23.3% feeling extremely confident and 29% feeling fairly confident. Satisfaction with the support and resources for antibiogram utilization in Saudi Arabia was neutral for 36.4% of the participants, while 30.1% were satisfied.

## 4. Discussion

This cross-sectional study provides valuable insights into the availability, utilization, and perceived impact of antibiograms among healthcare providers in Saudi Arabia. The findings highlight both the progress made and the challenges that remain in optimizing the use of antibiograms to guide antibiotic prescribing practices.

This study found that the majority of participants (62.2%) reported the availability of antibiograms at their healthcare facilities, indicating a relatively high level of access to this critical resource. However, the remaining participants either reported unavailability (16.6%) or were unsure (21.2%), suggesting that there is still room for improvement in ensuring consistent access to antibiograms across all healthcare settings.

The key barriers to antibiogram availability identified in this study were a lack of expertise, technological infrastructure, and funding. These findings align with previous research that has emphasized the importance of building institutional capacity, investing in necessary resources, and securing adequate funding to support tackling antimicrobial resistance, including establishing and maintaining robust antibiogram systems [12]. Addressing these challenges would be crucial for enhancing the accessibility of antibiograms throughout the Saudi healthcare system.

Among those with access to antibiograms, the majority (84.1%) reported utilizing them in their clinical practice. This high adoption rate reflects a growing awareness among healthcare professionals about the need for evidence-based approaches to antibiotic prescribing. By tailoring treatments based on local antibiogram data, clinicians can make more informed decisions, potentially improving patient outcomes and combating antibiotic resistance. However, the frequency of referring to antibiograms varied, with only 21.6% consulting them daily and a significant proportion (33.5%) doing so monthly or annually. This underscores the need to promote more consistent and frequent use of antibiograms to optimize their impact on antibiotic stewardship.

This study also identified several factors that influence the decision to consult antibiograms, including the patient’s clinical presentation and the failure of previously administered antibiotics. This aligns with the recommended practice of using antibiograms as a complementary tool to clinical judgment, considering both patient-specific factors and local resistance patterns [13].

Regarding the perceived impact of antibiograms, the majority of participants observed positive changes in antibiotic prescription practices and believed that antibiograms could contribute to reduced inappropriate antibiotic use, shorter patient length of stay, and cost savings. However, the perceived effectiveness of antibiograms in reducing antibiotic resistance was more varied, with only 27.8% finding them moderately effective. This suggests that while antibiograms are viewed as valuable tools, healthcare providers may not fully recognize their potential to address the broader challenge of antimicrobial resistance. Ongoing education and training on the role of antibiograms in antimicrobial stewardship could help strengthen this understanding. A systematic review by Khatri et al. [5] found that antibiograms, when used as part of a multifaceted intervention, can improve antibiotic use, appropriateness, and costs. However, Schulz et al. [14] cautioned that the relationship between hospital antibiotic use and resistance, as reflected in the antibiogram, is complex and inconsistent. Dikkatwar [15] emphasized the role of antibiograms in fighting antimicrobial resistance and highlighted the need for their development.

This study also highlighted the need for continuous efforts to improve antibiogram data quality, timeliness, and interpretability, as these were identified as obstacles to effective utilization. Additionally, the finding that only 40.9% of participants had received training on antibiogram use underscores the importance of investing in educational initiatives to enhance the competency of healthcare providers in interpreting and applying antibiogram data.

This study has several limitations that should be considered when interpreting the results. First, the cross-sectional design captures data at a single point in time, which may not fully reflect changes in antibiotic prescribing practices over longer periods. Longitudinal studies would provide a more comprehensive understanding of the impact of antibiogram utilization on antibiotic resistance and prescribing behaviors. Second, the reliance on self-reported data from healthcare providers introduces the potential for response bias. Participants may have overestimated their use of antibiograms or their effectiveness in guiding antibiotic prescribing. Additionally, the voluntary nature of survey participation could result in selection bias, with those more interested in antimicrobial stewardship being more likely to respond. Third, this study was conducted in various healthcare settings across Saudi Arabia, but the sample may not be fully representative of all healthcare providers in the country. Regional differences in healthcare infrastructure and practices may influence the generalizability of the findings. Fourth, while the survey underwent content and face validity processes, the potential for misinterpretation of survey questions by participants cannot be entirely ruled out. This may have affected the accuracy of the responses regarding antibiogram availability and utilization. Fifth, this study did not assess the actual clinical outcomes associated with antibiogram use, such as patient recovery rates or the incidence of antibiotic-resistant infections. Future research should aim to link antibiogram utilization directly to clinical outcomes to better evaluate their effectiveness. Lastly, this study identified several barriers to antibiogram availability and utilization, such as lack of expertise and technological infrastructure. However, it did not explore in depth the specific strategies or interventions that could address these barriers. Further research is needed to develop and test targeted interventions that could enhance the accessibility and effectiveness of antibiograms in diverse healthcare settings. Despite these limitations, this study provides valuable insights into the current state of antibiogram availability and use in Saudi Arabia and highlights important areas for improvement in antimicrobial stewardship practices.

## 5. Conclusions

This study provides a comprehensive assessment of the availability, utilization, and perceived impact of antibiograms among healthcare providers in Saudi Arabia. The findings suggest that while progress has been made, there are still opportunities to enhance the accessibility, frequency of use, and overall impact of antibiograms in guiding antibiotic prescribing practices and contributing to antimicrobial stewardship efforts. Addressing the identified barriers and implementing targeted interventions to improve antibiogram utilization and interpretation could further strengthen the role of this critical tool in optimizing antibiotic use and improving patient outcomes in the Saudi healthcare system.

## Figures and Tables

**Figure 1 microorganisms-12-01444-f001:**
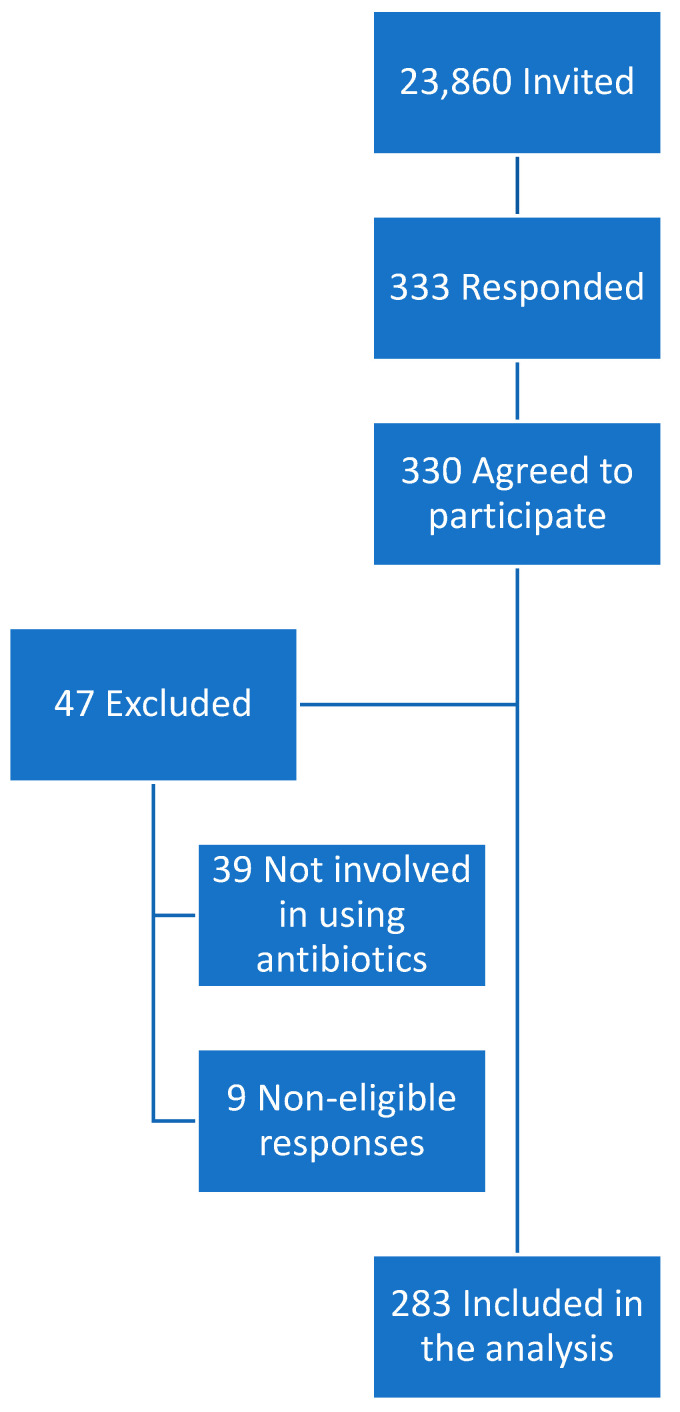
Flow diagram of the study participation.

**Table 1 microorganisms-12-01444-t001:** Demographic information of the study’s participants (*n* = 283).

Item	Value
Age, Years, *n* (%)	
Less than 30	92 (32.5)
31–40	137 (48.4)
41–50	48 (17)
51–60	6 (2.1)
Gender, *n* (%)	
Female	155 (54.8)
Nationality, *n* (%)	
Saudi	199 (70.3)
Non-Saudi	84 (29.7)
Geographic Region, *n* (%)	
Eastern Region	128 (45.2)
Central Region	80 (28.3)
Western Region	42 (14.8)
Southern Region	19 (6.7)
Northern Region	14 (4.9)
Job Title, *n* (%)	
Physician	114 (40.3)
Dentist	17 (6)
Pharmacist	95 (33.6)
Nurse	57 (20.1)
Specialty, *n* (%)	
Non-specialized	117 (41.3)
Family medicine	34 (12)
Infectious diseases	10 (3.5)
Oncology	6 (2.1)
Emergency medicine	13 (4.6)
Cardiology	9 (3.2)
Nephrology	8 (2.8)
Pediatrics	9 (3.2)
Other *	77 (27.2)
Job Sector, *n* (%)	
Ministry of Health hospitals	126 (44.5)
Private hospitals	111 (39.2)
University hospitals	16 (5.7)
Ministry of Defense hospitals	5 (1.8)
Ministry of National Guard hospitals	19 (6.7)
King Faisal Specialist Hospital and Research Centre	6 (2.1)
Years of Experience, *n* (%)	
Less than one year	24 (8.5)
1–5 years	72 (25.4)
6–10 years	82 (29)
11–20 years	77 (27.2)
More than 20 years	28 (9.9)
Type of Healthcare Facility, *n* (%)	
Primary care centers	14 (4.9)
General hospitals	160 (56.5)
Specialized hospitals	84 (29.7)
Outpatient clinic	19 (6.7)
Long-term care facility	5 (1.8)
Other	1 (0.4)

* Other specialties include critical care, dermatology, endocrinology, gastroenterology, surgery, and gynecology.

**Table 2 microorganisms-12-01444-t002:** Evaluation of antibiogram availability (*n* = 283).

Item	Value
Antibiogram availability at the healthcare facility, *n* (%)	
Available	176 (62.2)
Unavailable	47 (16.6)
Unknown	60 (21.2)
Reasons for the unavailability or the lack of updates on the antibiogram, *n* (%) ^a,b^	
Lack of funding	21 (44.7)
Lack of expertise	29 (61.7)
Lack of technology infrastructure and resources	27 (57.4)
Not needed	8 (17)
Facility’s plan to create its antibiogram in the future, *n* (%) ^a^	
Yes	5 (10.6)
No	10 (21.3)
Unknown	32 (68)
Reasons for the unawareness of the existence of the antibiogram, *n* (%) ^c^	
Recent employment at the facility	21 (38.2)
Lack of belief on its importance	23 (41.8)
Other	11 (20)
Accessibility of the antibiogram, *n* (%) ^d^	
Very easy	72 (41.1)
Moderately easy	73 (41.7)
Difficult	29 (16.6)
Very difficult	1 (0.6)

^a^ The data in this item represent the 47 participants who stated that the antibiogram is not available at their facility. ^b^ This question was formulated as a multiple-choice multiple response (MCMR) question. Therefore, each choice item of the question was calculated as a stand-alone response out of the total responses (i.e., adding the response percentages of all the items will add up to more than 100). ^c^ The data in this item represent the 60 participants who stated that they are unaware of the antibiogram availability at their facility (with five responses missing). ^d^ The data in this item represent the 176 participants who stated that the antibiogram is available at their facility (with one response missing).

**Table 3 microorganisms-12-01444-t003:** Evaluation of antibiogram utilization (*n* = 176).

Item	Value
Utilization of the antibiogram in the healthcare practice, *n* (%)	
Yes	148 (84.1)
No	28 (15.9)
Frequency of referral to an antibiogram when prescribing antibiotics, *n* (%)	
Daily	38 (21.6)
Weekly	45 (25.6)
Monthly	59 (33.5)
Yearly	34 (19.3)
Factors influencing the decision to consult an antibiogram before prescribing antibiotics, *n* (%) ^a^	
Patient’s age	55 (31.3)
Patient’s clinical presentation	120 (68.2)
Failure to respond to a previously given antibiotic	115 (65.3)
Type of bacterial infection	88 (50)
Other	9 (5.1)
Participants’ frequency of updating their knowledge about new antibiotics and their resistance patterns through antibiograms, *n* (%)	
Always	41 (23.3)
Sometimes	59 (33.5)
Rarely	56 (31.8)
Never	20 (11.4)
Frequency of updating the antibiogram by the healthcare facility, *n* (%)	
Every six months	39 (22.2)
Every year	105 (59.6)
Every two year	22 (12.5)
Every three years or more	10 (5.7)
Possible obstacles that may hinder the use of antibiograms, *n* (%) ^a^	
Data quality	99 (56.3)
Delays in data updates or a lack of real-time data	66 (37.5)
Difficulty in data interpretation	65 (36.9)
Limited awareness about antibiogram availability and its importance	127 (72.2)
Other	2 (1.1)
Receiving training or guidance on utilizing antibiograms in the clinical practice, *n* (%)	
Yes	72 (40.9)
No	104 (59.1)
Receiving feedback on antibiotic prescribing practices based on antibiogram data, *n* (%)	
Yes	82 (46.6)
No	94 (53.4)
Participants’ likeliness of incorporating antibiogram recommendations into antibiotic prescribing decisions, *n* (%)	
Very unlikely	12 (6.8)
Unlikely	27 (15.3)
Neutral	51 (29)
Likely	51 (29)
Very likely	35 (19.9)
Participants’ rating of the quality and accuracy of the available antibiograms at the facility, *n* (%)	
Very poor	2 (1.1)
Poor	31 (17.6)
Fair	62 (35.2)
Good	52 (29.5)
Very good	29 (16.5)
Participants’ confidence in interpreting and applying the information in an antibiogram, *n* (%)	
Not confident at all	9 (5.1)
Slightly confident	19 (10.8)
Somewhat confident	56 (31.8)
Fairly confident	51 (29)
Extremely confident	41 (23.3)
Participants’ satisfaction with the support and resources available for antibiogram utilization in Saudi Arabia, *n* (%)	
Very dissatisfied	5 (2.8)
Dissatisfied	33 (18.8)
Neutral	64 (36.4)
Satisfied	53 (30.1)
Very satisfied	21 (11.9)

^a^ This question was formulated as a multiple-choice multiple response (MCMR) question. Therefore, each choice item of the question was calculated as a stand-alone response out of the total responses (i.e., adding the response percentages of all the items will add up to more than 100).

**Table 4 microorganisms-12-01444-t004:** Participants’ perception of the impact of antibiograms on antibiotic use (*n* = 176).

Item	Value
Observation of any changes in antibiotic prescribing practices since the introduction of the antibiogram, *n* (%)	
Yes	121 (68.8)
No	55 (31.2)
Effectiveness of utilizing antibiograms in reducing inappropriate antibiotic use and antibiotic resistance, *n* (%)	
Very ineffective	4 (2.3)
Ineffective	18 (10.2)
Moderately effective	49 (27.8)
Effective	58 (33)
Very effective	47 (26.7)
The availability and utilization of antibiograms can shorten patients’ length of stay, *n* (%)	
Yes	153 (86.9)
No	23 (13.1)
The availability and utilization of antibiograms can contribute to improved patient outcomes or prognosis, *n* (%)	
Yes	149 (84.7)
No	27 (15.3)
The availability and utilization of antibiograms lead to cost savings in healthcare settings, *n* (%)	
Yes	150 (85.2)
No	26 (14.8)
The antibiogram data are incorporated into the hospital’s empirical treatment guidelines, *n* (%) ^a^	
Yes	95 (54.3)
No	37 (21.1)
There are no empirical treatment guidelines	43 (24.6)
The overall impact of antibiogram availability in the healthcare setting, *n* (%)	
Very poor	1 (0.6)
Poor	34 (19.3)
Acceptable	52 (29.5)
Good	53 (30.1)
Very good	36 (20.5)

^a^ One response was missing for this item.

## Data Availability

The data presented in this study are available upon request from the corresponding author due to restrictions on sharing sensitive or personal information.

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
