# Peer review of "Evaluation of the Impact of Antibiogram Availability and Utilization on Antibiotic Use among Healthcare Providers in Saudi Arabia: A Cross-Sectional Study"

_microorganisms, 2024, doi:10.3390/microorganisms12071444_

Round 1

Reviewer 1 Report

Comments and Suggestions for Authors

Comments and Suggestions:

Summary section:

1. The last sentence of this section should be deleted.

2. Keywords must be MeSH terms

Introduction Section:

3. Due to the incorporation of molecular platforms for the identification of resistance genes, the advantage of the antibiogram over these platforms should be mentioned in the introduction.

4. At least one paragraph should be added on the usefulness of antibiograms in clinical practice.

5. A paragraph should be added indicating why such a diverse population of health professionals and not just doctors has been selected.

Materials and methods section:

6. Explain the face validity process

Results section:

7. Restructure section 3.2 for better understanding

Conclusions section:

8. Restructure according to the objectives of the study

Comments on the Quality of English Language

Minor editing of English language required

Reviewer 2 Report

Comments and Suggestions for Authors

Dear Authors,

Interesting study that highlights the need for a better implementation of the use of the antibiogram in antibiotherapy, but also its usefulness, in the geographical area of interest of the study. Thus, the concern of researchers is all the more salutary, since antibiotic resistance has a major impact on global health, and the improper use of antibiotics must be limited by all possible means. And this preoccupation can represent a possible strategy in the most targeted therapy in current antibiotic therapy.

I suggest some small modifications or corrections in the manuscript:

The main purpose of the study remains to bring to the attention of specialized staff the importance of the antibiogram in making strategic treatment decisions, but at the same time it brings to attention the urgent need to limit the emergence and spread of microbial resistance to antibiotics, supported of course by citations from the specialized literature.

The number of participants in this transversal study, as presented, is 283 included participants, a number that can be considered acceptable for eligibility, and in table number 1, 77 participants with other professions are included, which represents a significant percentage (27%) . Considering the specification initially given on the proportion of the medical specialty staff that prevails, I suggest the authors to specify the profession of these 27%, if the medical professorial area is maintained or not.

The statistical analysis of the study highlights once again the importance of using the antibiogram in approaching an antibiotic treatment strategy, considering that the majority of participants (84.1%) reported that they used them in their clinical practice.

Round 2

Reviewer 1 Report

Comments and Suggestions for Authors

The authors have responded to my comments and suggestions to the best of their ability. I have no further comments on this.  

Comments on the Quality of English Language

Minor editing of English language required